# Distribution Characteristics and Risk Assessment of Agricultural Land Use Non-Point Source Pollution in Typical Biofuel Ethanol Planting Areas

**DOI:** 10.3390/ijerph19031394

**Published:** 2022-01-26

**Authors:** Guannan Cui, Yanfeng Liu, Pengfei Wang, Xinyu Bai, Haitao Wang, Yiming Xu, Meiqiong Yang, Liming Dong

**Affiliations:** 1School of Ecology and Environment, Beijing Technology and Business University, Beijing 100048, China; cuiguannan@btbu.edu.cn (G.C.); 13568264129@163.com (X.B.); wanghaitao008@st.btbu.edu.cn (H.W.); xuyiming@btbu.edu.cn (Y.X.); 2State Environmental Protection Key Laboratory of Food Chain Pollution Control, Beijing Technology and Business University, Beijing 100048, China; 3Key Laboratory of Cleaner Production and Integrated Resource Utilization of China National Light Industry, Beijing Technology and Business University, Beijing 100048, China; 4WonderFlow Environmental Science and Technology (Beijing) Co., Ltd., Beijing 100192, China; ziyuemingtian@163.com; 5National Engineering Laboratory, Lake Pollution Control and Ecological Restoration, Chinese Research Academy of Environmental Sciences, Beijing 100012, China; wangpf01@craes.org.cn; 6State Environment Protection Key Laboratory, Lake Pollution Control and Ecological Restoration, Chinese Research Academy of Environmental Sciences, Beijing 100012, China; 7Agricultural and Rural Center of Baisha Town, Guiping 537221, China; ymq2163@163.com

**Keywords:** LUCC, non-point source pollution, energy policy, crop structures, nitrogen and phosphorus forms

## Abstract

Speeding up the promotion and application of bio-fuel ethanol was a national strategy in China, which in turn affected changes in the raw material planting structure. This study analyzed the distribution of nitrogen and phosphorus forms in water bodies and the soil of the typical maize and cassava fuel ethanol raw material planting areas. The results revealed that the maize planting area faced more serious TN and TP pollution. The river pollution was greatly affected by TN, TP, Ex-P and Fe/Al-P in soil, while soil TN and NO_3_^−^-N were the main factors influencing its counterpart. Furthermore, the risk assessment of soil nitrogen and phosphorus loss was carried out based on planting structures of crops. We investigated whether the water quality indexes or soil nitrogen and phosphorus loss risk assessment results showed that the Yujiang River stayed significantly less polluted. It was proven that the cassava planting area was more suitable for vigorously developing fuel ethanol. As for the high-risk areas, ecological agriculture promoting and fertilizer controlling measures were suggested. Under the change of the fuel-ethanol policy, this study could provide scientific support for the assessment of the impact of the Chinese national fuel ethanol policy on the water environment of the raw material planting area.

## 1. Introduction

Recently, growing population and urbanization have accelerated global energy demand. It is estimated that global energy consumption will increase by about 50% in the next 15 years [1]. The non-renewable nature of traditional fossil fuels and serious environmental problems caused by its overuse have become a great challenge to the world’s energy sustainability. It is urgent to seek green and clean renewable energy for sustainable development [2]. At present, biofuel is one of the alternatives to improve energy security and eco-friendly development, and has been favored by countries all over the world. Most countries issued relevant laws or policies on the promotion and application of biofuel ethanol, which represented a promising prospect of the fuel ethanol industry [3]. China’s fuel ethanol implementation plan has developed since the beginning of the 21st century and promoting the use of vehicle bioethanol gasoline showed an increasing trend year-by-year [4]. The alternatives of energy revolution would be conductive to realize carbon neutralization and fundamentally move towards the road of green development.

China’s fuel ethanol production mainly used maize and other food crops in the north areas and cassava in the south areas as raw materials. However, with the development and the increasing demand of biofuel ethanol, the problem of “competing with people for food” such as maize raw materials gradually became prominent. Compared with the crops mentioned above, cassava had incomparable advantages to become the main force in the raw materials because of its low production cost and high production capacity among similar non-grain crops [5]. Different planting structures had effects on surface water and soil to a great extent of the river basin and were essential factors for local develop planning.

Numerous studies discussed the nitrogen and phosphorus distribution characters of a typical river basin and the results played an important role in guiding the land use or cultivation management for local planning [6,7,8]. Frequent human activities such as land use changes or excessive chemical fertilizer applications could cause serious agricultural non-point source pollution [9]. A comprehensive understanding and proper evaluation of the risks caused by the main risk sources in the river basin to the ecological environment and human society could reduce their vulnerability in critical situations and provide a theoretical basis for the future risk management and governance [10]. Soil nitrogen and phosphorus loss was an important source and its loss risk assessment was the keynote for researchers [11,12,13] assessed the phosphorus loss risk as characteristic pollutants of agricultural soil in the Danjiangkou watershed in China. Universal Soil Loss Equation (USLE) was the basis and the most widely used prediction method of nitrogen and phosphorus loss in the world, because its loss factors were relatively easy to determine based on abundant experimental observation data [14,15,16] evaluated the risk of soil nitrogen and phosphorus loss by this method around the Yishu River Basin in China, and found that the risk in industrial developed areas was slightly higher than that in agricultural developed areas. NPS (non-point source) pollution risk assessment was established the in the Fuxian Lake Basin by Revised Universal Soil Loss Equation [17] expanding field data and combining soil erosion data. Most studies carried out the land use factors calculation based on the first classifications quoted in *Land Use Classification GBT21010-2017.* The planting structures of crops were not thoroughly considered in the past studies. In the future, the research on non-point source pollution would gradually transit to fine simulation and management at filed scales [18].

In this study, two typical fuel ethanol raw material planting areas were selected: maize planting area in Harbin section of the Hulan River Basin in Heilongjiang Province and cassava planting area in the Guiping section of the Yujiang River Basin in Guangxi Province. The research content was mainly divided into the following three tasks: (1) Occurrence characteristics of nitrogen and phosphorus in water and soil were revealed of maize and cassava growing areas through the analysis of river water quality and the spatial distribution. Combined with the multivariate statistical analysis method, the source of river non-point source pollution was analyzed. (2) The risk assessment of soil nitrogen and phosphorus loss was carried out based on planting structures of crops, which aimed to identify high loss risk areas for further risk control management. The results would provide scientific support and management suggestions for the assessment of the national fuel ethanol policy impacts on the water environment of the raw material planting area.

## 2. Materials and Methods

### 2.1. Overview of the Study Area

In this study, two typical biofuel ethanol raw material planting areas were selected:

#### 2.1.1. Maize Planting Area (Harbin Section of Hulan River Basin)

As shown in Figure 1, the study area is located in Hulan District, Harbin. The Hulan River is a tributary of Songhua River system, which originates from Shulan City in Jilin Province. The main stream has a total length of 128 km and a drainage area of 1502 km^2^. This research takes place in the area where the Hulan River flows from northwest to southeast into the Songhua River, with a total length of about 40 km. The whole study area dominates about 1109.6 km^2^ which is located at 126°42′55″ E~126°26′01″ E, 45°55′56″ N~46°06′28″ E. The terrain is almost dominated by plain. The Hulan River experiences a wet season from June to September every year, and the flow accounts for 60%~80% of the annual flow with water source coming mainly from precipitation. 

In order to study the characteristics of non-point source pollution in different periods, two sample sets were taken in this area. The first set was in September 2020, during the wet season of Hulan River. The second was in November 2020, which was in the normal water period of the study area. The grid point distribution method was used for soil sampling, with a total of 109 sampling points and 17 water sampling points from upstream to downstream. 

The distribution of land use types of the Hulan River Basin was illustrated in Figure 2 (data source from the National Bureau of Soil Resources and interpretation for planting structures). Among them, cultivated land covered 72.4%, and irrigation land and construction land occupied 9.96% and 9.95%, respectively. The river area accounted for 5.07%. The three kinds of main crops were maize (41.68%), rice (30.31%) and soybeans (2.59%), among which maize accounted for the largest proportion. The region of meadow soil held the largest portion, accounting for more than 70% in the study area, followed by black soil. 

#### 2.1.2. Cassava Planting Area (Guiping Section of the Yujiang River Basin)

Guangxi Province is the largest cassava planting province in China. The selected area is a typical raw material planting area for cassava fuel ethanol. Yujiang River is the largest tributary of the Xijiang River System in the Pearl River Basin. The Guiping section of Yujiang River Basin has a total drainage area of 411.22 km^2^. This research takes place in the area where the Yujiang River flows from southwest to northeast, with a total length of about 45 km (Figure 3). The wet season in Guangxi is from May to October, and the mean flow season is from November to April of the next year. The sampling time in this area was November 2020 and June 2021. The grid point distribution method was used for sampling, with a total of 53 soil sampling points and 16 water sampling points.

The distribution of land use types in cassava planting area was shown in Figure 4, in which the occupied areas of cultivated land and irrigated farmland were 41.65% and 40.01%, respectively. The construction land accounted for 7.79%, the river area was about 4.89%. In addition to cassava (2.31%), there were maize (2.97%) and rice (43.70%) in this area. The soil types in cassava planting area included rice soil, purple lime soil, yellow lateritic red soil, etc., of which acid purple soil and yellow lateritic red soil occupied most of the area. 

### 2.2. Experimental Method

The soil samples were taken from the surface (depth of 20 cm) and put into a 4 °C incubator and then carried back to the laboratory. After air-drying the soil samples for 2~3 weeks and grinding, a 50 g soil sample was sieved through 100 mesh for determination. The water samples were collected in the middle of river course from the boat and was placed in 250 mL polyethylene plastic bottles moistened with river water in an incubator at 4 °C. For those requiring nitrogen and phosphorus detection, sulfuric acid was used to acidify to pH < 2.

#### 2.2.1. Determination of Soil Nitrogen and Phosphorus Forms

The determination methods for water and soil nitrogen and phosphorus forms were summarized in Table 1.

#### 2.2.2. Loss Risk Assessment by Nitrogen and Phosphorus Index Method

Nitrogen and phosphorus index method is an evaluation method to estimate the risk of soil nitrogen and phosphorus loss. The factors to be collected in this method are mainly divided into source factors and migration factors [19]. According to the role of factors in the loss process that N and P play, corresponding weights are given to calculate the nitrogen and phosphorus index [20]. The calculation method is based on Equation (1) as follows:(1)$$I=[\sum (S_{i}\times W_{i})]\times \prod (T_{j}\times W_{j})$$
where I is nitrogen or phosphorus index; *S_i_* is the grade value corresponding to the source factor evaluation index I; *W_i_* is the weight corresponding to index I; *T_j_* is the grade value corresponding to the evaluation index *j* of migration factor; *W_j_* is the weight corresponding to the migration factor index *j*.

The weight and classification of each factor of nitrogen and phosphorus index are shown in Table 2 and Table 3.

(1)Source factor calculation method

The source factors mainly reflect the input of TN, TP and nitrogen and phosphorus fertilizer in the soil of the study area. The content of TN and TP in soil, the application amount, time and patterns of nitrogen and phosphorus fertilizer are the main source factors. 

Fertilization amount: Chemical fertilizer are an important source of TN and TP in agricultural soil. The fertilization amount used in this study was looked for from the data in the national collection of cost–benefit data of agricultural products in 2019, which had been confirmed through field investigation.

Application time: The application time of fertilizer and rainfall cycle have an important impact on the risk of loss. Fertilization in rainy season will inevitably bring more serious non-point source pollution. The application time factor in this study was obtained through field investigation.

Fertilization patterns: Common fertilization patterns include surface fertilization, sowing, burying and foliar spraying. It is generally believed that shallow surface fertilization will bring more serious loss [20]. In this study, fertilization methods and factors were obtained through field investigation.

(2)Migration factor calculation method
(i)Soil erosion

Universal Soil Loss Equation (USLE) is commonly used to estimate the amount of soil erosion, and the expression is:A = R × K × L × S × C × P(2)
where A is the amount of soil erosion (t (hm^2^·a)^−1^); R is rainfall erosion factor; K is soil erodibility factor; L and S are slope and slope length factors; C is vegetation cover and management factor; P is the water and soil conservation factor

(ii)Rainfall erosion

Rainfall erosion factor represents the potential capacity of soil erosion caused by rainfall, which is related to rainfall duration and intensity. At present, the most common algorithm of rainfall erosion factor is the simple estimation method of monthly rainfall.
(3)$$R=\alpha F_{F}^{\beta }$$
(4)$$F_{F}=\frac{\sum_{i=1}^{N}[(\sum_{j=1}^{12}P^{2})/[(\sum_{j=1}^{12}P)]}{N}$$
where F_F_ is annual rainfall erosion factor (MJ·mm·(hm^2^·a)^−1^); P is the rainfall in the j month of the year I (mm); N is the number of years counted for calculating rainfall (a); R is the annual average rainfall erosion factor (MJ·mm (hm^2^·a)^−1^); α and β are model parameters.

(iii)Soil erodibility

The value of soil erodibility factor K is mainly related to soil particle size.
(5)$$K=[2.1\times 10^{-4}\times {\left(N_{1}\times N_{2}\right)}^{1.14}(12-O_{M})+3.25(S-2)+2.5(P-3)]/100$$
where N_1_ = (very fine sand + silt)%, N_2_ = (100 clay)%; O_M_ is the percentage of organic matter content; S is the grade coefficient of soil structure; P is the coefficient of soil permeability grade.


(iv)Terrain


Terrain factors S and L are slope and slope length, respectively. Generally, the terrain factors can be obtained by GIS, but when the study area data cannot be available, it can be estimated according to the formula as follows:(6)$$L\times S={\left(0.45L\right)}^{\alpha }(65.41\sin^{2}\beta +4.65\sin\beta +0.065)$$
where L is slope length (m); β is slope inclination; S is the slope; α is the slope length index. When s ≥ 5%, take α= 0.5, when 3.5% < s < 4.5%, take α = 0.4. When 1% < s < 3%, take α = 0.3. When s < 1%, take a = 0.2. The Hulan River Basin is located in a plains area, so s < 1% is taken uniformly, α = 0.2.

(v)Vegetation cover and management, soil and water conservation and distance

Vegetation coverage and management factors are usually set to C = 0.001~1.0, where 1.0 means no vegetation coverage and 0.001 means complete coverage. The vegetation coverage in this area is high, and the crops are mainly rice and maize.

Soil and water conservation factor P is the most difficult data to obtain in soil erosion calculation, and the acquisition method is not unified. The method used in this paper is the empirical formula method [21] as shown in Equation (7).
(7)P=0.2+0.03S
where P is soil and water conservation factor; S is the slope (%).

The distance factor is determined by the distance between the sampling point and the river.

The risk assessment was based on the soil nitrogen and phosphorus forms data gained from filed observation which was shown in Appendix A.

## 3. Results and Discussion

### 3.1. Water Quality Characteristics of Typical Fuel Ethanol Raw Material Planting Area

#### 3.1.1. Water Quality Characteristics in Maize Planting Area (Hulan River Harbin Section)

There were 17 water sampling points in the maize planting area of the Hulan River’s Harbin section. The river flowed from northwest to southeast and the sample points were marked as P1 to P17 in sequence. The water quality characters of the two phases are shown in Figure 5. 

Except P16, the TN concentrations in other sampling points exceeded the Chinese national class V water standard (>0.2 mg L^−1^) in September. In November, except P1 the rest sample points displayed the same characters as those of September. To sum up, the TN concentrations of the Hulan River water body seriously exceed the water quality standard, and the TN pollution degree in the normal water period was higher than that in the wet water period. As to the NH_4_^+^-N concentrations, sampling points P8~P15 and P17 met the class I standard (<0.15 mg L^−1^), and P1~P7 met the class II standard (0.15~0.5 mg L^−1^) in September. In November, P5 met the class III standard while the rest met the class II standard, and the NH_4_^+^-N concentrations in November were generally higher than those in September. 

As for the TP concentrations, P2, P4 and P6 met the class IV standard (0.2~0.5 mg L^−1^), and the rest samples met the class III water standard (0.1~0.2 mg L^−1^). In November, P1 was in class V while the rest ones met the class III standard. TP concentrations in Hulan River water body exceeded the water quality standard. In addition, except P2 the TP concentrations of all sampling points in September were higher than those in November, which indicated contrary patterns compared with the TN results. 

The COD concentrations of the Hulan River water samples in September and November were 23.6~55.3 mg/L and 26.1~46.5 mg/L, respectively. According to the environmental quality standard for surface water (GB3838−2002), the water samples in September were almost in class IV (30~40 mg/L) or class V (>40 mg/L), except point W17, which was in class III (20~30 mg/L). In November, W12 and W14~16 met the class III water standard, and the rest were situated in class IV or class V. Comparing the two sets of water samples, it was found that the COD concentrations of W1, W3, W6, W15, W16 and W17 in November were higher than those in September with the rest samples showed the opposite trends. It could be seen that the COD concentrations of some sampling points in the Hulan River exceeded the standard of water quality, and there were no great differences between the two phases. TOC contents were 9.00~17.10 mg/L in September and 5.87~19.40 mg/L in November. W1, W5 and W14~W17 had higher TOC concentrations in November while the other points showed conversely.

In conclusion, according to the characteristics of water quality sampled in September and November, it was found that the TN concentrations in the Hulan River Basin seriously exceeded the water quality standard. TN concentrations in water body of wet season were lower than those of mean flow season, which were similar to the research results of Xie et al. [22]. This situation may be due to the decrease in water volume resulting in higher concentrations. TP concentrations were on the contrary, because a large part of TP contents in the river came from the release of river bottom sediments. The endogenous release was greatly affected by temperature which a sensitive factor in Harbin Province. The higher the temperature was, the faster the TP released. The obviously lower temperature in November slowed down the release rate of TP contents. The concentrations of TN, TP, NH_4_^+^-N and NO_3_^−^-N in the Hulan River decreased along with the flow direction while TOC concentrations performed just the opposite. The land use type around the upper half of river was mainly agricultural cultivated land, while the lower half was almost urban construction land. It revealed nitrogen and phosphorus pollutants mainly came from agricultural pollution, while TOC was more likely to come from urban pollution sources. In the light of the investigation results of pollution sources around the Harbin section of the Hulan River, there were six centralized sewage outlets in the study area. All of them were located in the downstream, which stood a good chance to provide higher TOC for the Hulan River.

#### 3.1.2. Water Quality Characteristics in Cassava Planting Area (Yujiang River Guiping Section)

There were 16 water sampling points in the maize planting area of Yujiang River Guiping section. The river flowed from southwest to northeast and the sample points were marked as Y1 to Y16 in sequence. The water quality characters of the two phases were shown in Figure 6.

The TN concentration in the Yujiang river was 2.01~2.93 mg L^−1^ in November, and the TN concentration in water quality exceeded the class V water standard of surface water in China (2.0 mg L^−1^). This problem was alleviated in summer, but the TN concentration in Yujiang river still met the class III water standard (1.0 mg L^−1^). It could be seen that the TN concentration in Yujiang seriously exceeds the standard.

The concentration of NH_4_^+^-N in Yujiang in November was 0.088~0.255 mg L^−1^. Y2, Y5 and Y15 met China’s class II surface water standard (0.5 mg L^−1^), and other sampling points met China’s class I surface water standard (0.15 mg L^−1^). Correspondingly, in June, the overall ammonia nitrogen level was slightly higher than that in November. In terms of nitrate nitrogen, the concentration of NO_3_^−^-N in Yujiang was 0.5~0.8 mg L^−1^ in June and close to 1.2 mg L^−1^ in November. The concentrations at each monitoring point in November were close and significantly higher than those in June.

The TP concentration in the Yujiang River was 0.06~0.15 mg L^−1^ in November. The data showed that except Y1 and Y2 sampling points met the class II standard of surface water (0.1 mg L^−1^), other points only met the class III standard of surface water (0.2 mg L^−1^). The TP concentration reached the standard in summer and exceeded the standard in winter.

The COD concentration was 4~5 mg/L in November. In June, the variation range of COD concentration in water body was large (2~9 mg/L). COD concentration in the water body met the class I water standard of China’s surface water in winter and summer (the COD standard of class I water is the same as that of class II water, both 15 mg/L), which shows that the COD of the Yujiang River did not exceed the standard.

In conclusion, combined with the water quality characteristics of the Yujiang River in June and November, the data revealed that although the total nitrogen content of the Yujiang River exceeded the standard seriously, the other indicators met the water quality standards of the region or only slightly exceed the standard. Compared with November, the content of nitrogen and phosphorus in the Yujiang River in June was lower, which was speculated to be related to the large precipitation in summer. Therefore, the water quality of the Yujiang River was at better state than that of the Hulan River.

As for the two river basins, the COD concentrations of the Yujiang River were significantly lower than those of the Hulan River with a large difference between them. The average COD concentration value of the Yujiang River was 4.5 mg/L, which fulfilled the class I standard requirement. Additionally, that of the Hulan River was 37.388 mg/L, which only met the class IV standard. The average TN concentration value in the Hulan River was 2.904 mg/L, which was higher than that of the Yujiang River Basin (2.356 mg/L). The same principles appeared in the TP situation with the data 0.126mg/L of the Yujiang River and 0.233 mg/L of the Hulan River. Concerning the NO_3_^−^-N and NH_4_^+^-N circumstances, the observations presented quite distinct results. The average NO_3_^−^-N concentration value of the Yujiang River (1.231 mg/L) was significantly higher than that of the Hulan River (0.771 mg/L), while the NH_4_^+^-N values of the two basins stayed closely (0.143 mg/L of the Yujiang River and 0.146 mg/L of the Hulan River). Overall, the water quality of the Yujiang River of cassava planting area in the south was obviously better and steadier than that of the Hulan River of maize planting area in the north. The data from the north research points displayed more marked differences among each other and emerged more outlier data than the south area. 

### 3.2. Pollution Source Analysis of Water Quality

The principal component analysis of water quality indexes NO_3_^−^-N, NH_4_^+^-N, TN, TP, COD, TOC and turbidity was conducted in the Hulan River Basin, and the results were shown in Table 4.

Three principal components were extracted from the principal component analysis, and the total pollution contribution rate reached 81.449%. The high load factor (>0.500) was picked from the three principal components. The high loading factors in principal component 1 were NH_4_^+^-N, TN and turbidity, indicating that these three pollutants had homology. The farmland area of maize planting area reached more than 80%. The non-point source pollution caused by the loss of nitrogen and phosphorus in farmland soil was likely to be the main factor of water pollution. Turbidity in surface water mainly originated from soil particles, therefore principal component 1 could come from soil and water loss. NH_4_^+^-N contents were easily adsorbed on soil particles in surface runoff which also became the main source of TN washed with soil and water loss. Principal component 2 contained NO_3_^−^-N and TP. As TP usually entered the river course in granular form and was released after settling to the sediment principal component 2 could come from endogenous pollution of river bottom sediments. The concentration of NO_3_^−^-N would have an important impact on the release of phosphorus in sediments. Nitrate could reduce sediment phosphorus release through oxidation and promote phosphorus release by stimulating phytoplankton growth. Alkaline phosphatase secreted by phytoplankton explains the phosphorus release. The effects of nitrate loading on sediment phosphorus release are dose-dependent [23], which made NO_3_^−^-N also become a high loading factor. The high load factors in principal component 3 mainly included TOC and COD. As the land use type around the lower half of the Hulan River was mainly urban construction land, it might be a carbon pollution source caused by human activities.

As shown in Table 5, three principal components were extracted from the principal component analysis of water quality indicators of the Yujiang River, and the contribution rate of total variance was 75.279%. It was lower than the variance contribution rate of the Hulan River water quality that indicated the factors were more dispersed and the characteristics of water pollution sources were not as significant as the Hulan River. The high load factors of principal component 1 were NO_3_^−^-N, TP and turbidity; the high load factors of principal component 2 were TN and TP; and those of principal component 3 were NH_4_^+^-N and turbidity. It was inferred that there were three main pollution sources in the water body of the Yujiang River Basin. The farmland area around the river covered more than 80% comprising proximate even measure of area for dry farmland and irrigated farmland. Irrigated farmland was more prone to loss. Therefore, principal component 1 should come from non-point source pollution in irrigated farmland. Principle component 2 contributed NO_3_^−^-N and TP load to the water body. In addition to a large area of farmland, there were many villages nearby. Principle component 2 was probable to come from the discharge of rural domestic sources. Principal component 3 was composed mainly of NH_4_^+^-N and turbidity. Since the dry field dominated more than 40% of the research area, this could come from the soil loss of dry farmland. 

In conclusion, through the source analysis of water quality indicators of the Hulan River and the Yujiang River, it was found that nitrogen and phosphorus pollutants of the two regions were related to agricultural non-point source pollution to a great extent. The water quality of the Hulan River was in more obviously serious situation. Therefore, it was essential to carry out the form analysis of soil nitrogen and phosphorus and loss risk assessment around the watershed.

### 3.3. Spatial Distribution Characteristics of Soil Nitrogen and Phosphorus 

#### 3.3.1. Spatial Distribution Characteristics of Soil Nitrogen of Maize Planting Area and Cassava Planting Area

(1)Spatial distribution characteristics of TN

The TN content distributions of sampling points in maize and cassava planting areas from the crop growth stage were demonstrated in Figure 7. Interpolation analysis was an unbiased optimal estimation of regional variables in a finite region [24]. Some scholars have compared several interpolation methods and found that the analysis process of the Kriging method was relatively simple with relatively reliable results [25]. The TN distribution map was calculated by Kriging interpolation method by GIS-ArcMap10.3 and the grid size were 100 × 100 m. The TN content in maize planting area ranged from 10 to 4240 mg kg^−1^, with an average value of 1924.429 mg kg^−1^. In accordance with the national soil abundance classification standard [26], the average TN content in maize planting area soil achieved the rich level. According to the TN abundance level of the two study areas, the classification was divided into five categories from low to high: <1500 mg kg^−1^ (lower), 1500~2000 mg kg^−1^ (low), 2000~2300 mg kg^−1^ (medium), 2300~2500 mg kg^−1^ (high) and >2500 mg kg^−1^ (higher). The area proportion was 15.6%, 52.1%, 21.5%, 8.4% and 2.4%, respectively. The TN content of soil near the river course was significantly lower than that far away from the river course due to the more vulnerable soil texture to erosion in the nearby river area. The lowest TN content area was near the estuary in the whole region.

The TN content in cassava planting area was from 310 mg kg^−1^ to 3300 mg kg^−1^, with an average value of 1620.181 mg kg^−1^. The average content also achieved rich level. The TN content distribution with lower, low and medium property accounted for 44.2%, 37.1% and 18.6% of the total, respectively. There were clear characters with lower content in the southwest while higher content in the northernmost. Comparing the TN content of the two study areas, the average TN content in cassava planting area was relatively lower. The soil area with low TN content was 28.6% more than that in maize planting area. This could be related to the fact that the local cassava planting did not need topdressing nitrogen fertilizer that consequently affected the water quality difference in the south or north typical bioethanol raw material planting areas.

(2)Spatial distribution characteristics of NH_4_^+^-N and NO_3_^−^-N

NH_4_^+^-N and NO_3_^−^-N in soil were the main forms of nitrogen loss in non-point source pollution. As shown in Figure 8, the average content of soil NH_4_^+^-N in maize and cassava planting area is 12.295 mg kg^−1^ and 7.114 mg kg^−1^, respectively. The average NO_3_^−^-N content value in soil was 13.917 mg kg^−1^ and 6.09 mg kg^−1^, respectively. Overall, the both indexes in maize planting area were higher than those in cassava planting area. On the one hand, it could be related to the need of nitrogen fertilizer application to maize growth, while cassava planting was usually no longer topdressing requirements. On the other hand, the soil in cassava planting area was mostly acidic which was more conducive to the NH_4_^+^-N stability than nitrification to produce NO_3_^−^-N. 

#### 3.3.2. Spatial Distribution Characteristics of Soil Phosphorus of Maize Planting Area and Cassava Planting Area

(1)Spatial distribution characteristics of TP

As shown in Figure 9, the average content of TP content in maize planting area (September) and cassava planting area (October) during crop growth period was 665.79 mg kg^−1^ and 619.67 mg kg^−1^, respectively. According to the classification of national TP abundance level [26], the TP content was divided into five grades from low to high as similar as the TN classification. The grade distribution in maize planting area basically covered five grades. The area at 600~800 mg kg^−1^ (medium) occupied the largest, up to 45.3% of the total. As to the cassava planting area, the TP content was basically 600~800 mg kg^−1^ (medium). 

(2)Spatial distribution characteristics of Ex-P, Fe/Al-P and Ca-P

As shown in Figure 10, Ex-P was the most soluble form of phosphorus in soil. After it was dissolved in surface runoff, it could enter the water body in the form of phosphate. The Ex-P content in maize planting area ranged from 2.45 mg kg^−1^ to 173.13 mg kg^−1^, with an average value of 19.11 mg kg^−1^. The Ex-P content in cassava planting area ranged from 0.78 mg kg^−1^ to 53.76 mg kg^−1^, with an average value of 9.93 mg kg^−1^. The maize planting area had slightly higher records, which might be caused by excessive fertilization. 

As shown in Figure 11, the Fe/Al-P content in maize planting area ranged from 41.00 mg kg^−1^ to 267.11 mg kg^−1^, with an average value of 104.51 mg kg^−1^. The Fe/Al-P content in cassava planting area ranged from 44.26 mg kg^−1^ to 725.38 mg kg^−1^, with an average value of 256.62 mg kg^−1^. The cassava planting area outnumbered its counterpart at Fe/Al-P content. The m000ain soil type in cassava planting area was lateritic red soil, which contained a very high Fe element resulting in the condition above.

As shown in Figure 12, the Ca-P content in maize planting area ranged from 21.82 mg kg^−1^ from 186.78 mg kg^−1^, with an average value of 74.159 mg kg^−1^. The Ca-P content in cassava planting area ranged from 0.884 mg kg^−1^ to 193.66 mg kg^−1^, with an average value of 59.306 mg kg^−1^. The difference was due to the low content of calcareous of acid soil type in cassava planting area in South China, which was not conducive to the formation of Ca-P.

In order to study the water quality response to spatial distribution of soil nitrogen and phosphorus, the redundancy analysis (RDA) in multivariate statistics was utilized in the study based on the observation data of sampling points. The spatial distribution characteristics of TP, Ex-P, Fe/Al-P and TN became the main factors affecting water quality in the maize planting area. In terms of correlation (the closer the distance, the better the correlation), Ex-P and TP assumed a good correlation with TP concentration in water. Additionally, NO_3_^−^-N showed a good correlation with NO_3_^−^-N and NH_4_^+^-N in water. 

As for the cassava planting area, the spatial distribution characteristics of TN and NO_3_^−^-N had become the main factors affecting water quality. Soil NO_3_^−^-N performed a good correlation with water TN, indicating that water TN might mainly come from NO_3_^−^-N in soil from slope flow.

### 3.4. Risk Assessment of Soil Nitrogen and Phosphorus Loss in Typical Fuel Ethanol Planting Areas

The potential risk assessment of soil nitrogen and phosphorus loss was very important for effective planning and control of soil non-point source pollution. For nitrogen and phosphorus loss, factors included surface soil texture, soil nitrogen and phosphorus retention/adsorption or buffer capacity, hydrology, soil erosion risk, land use and management and soil nutrient level.

Determination of source factors

The application of fertilizer was the main source of nitrogen and phosphorus in crop soil. The continuous excessive would aggravate the loss of soil nutrients. Table 6 listed the amount of nitrogen and phosphorus fertilizer for different crops in both areas. The data was achieved by *Compilation of national agricultural product cost–benefit data 2019* and field investigation.

Determination of migration factors

The algorithm of rainfall erosion factor was calculated by simple estimation method of monthly rainfall in this study. The rainfall erosion factor R in Harbin was 127.828, while that in Guiping was 188.653. The data of monthly rainfall was cited from *China Water Conservancy statistical yearbook 2017~2021*.

The value of soil erodibility factor K was mainly related to soil particle size. The average K value of each soil species in Heilongjiang and Guangxi Province was shown in Table 7 [27,28] after the literature research.

In this study, the sample points in maize and cassava planting areas were divided into three categories according to the slope, which were 0~2, 2~6 and 6~25, respectively. The slope length factor was obtained by grid calculation method in GIS after the slope classification. 

Based on the results of Li Bai’an [29] on the soil vegetation cover and management factors of dry land and rice field, the factors C of maize, soybean and rice were set to 0.24, 0.24 and 0.18, respectively. Soil and water conservation factor P was determined by Equation (7) combining the slope at each sampling point. The average P value of soil in maize planting area was 0.352 and that in cassava planting area was 0.230. The rating method was showed in Table 8 [20].

#### 3.4.1. Risk Assessment of Soil Nitrogen Loss

The risk maps of soil nitrogen loss of the two study areas were exhibited in Figure 13. Most of the maize planting areas are at medium and high nitrogen loss risk. The area of nitrogen index in 3~4 reached 39.7% of the total region, and the area of 4~5 reached 56.3%. The soil nitrogen index in cassava planting area was also mostly in the range of 3~4 and 4~5. The area with high nitrogen index was the middle and upper reaches and the West Bank, while the low nitrogen index appeared in the east bank of the middle and upper reaches as well as downstream. Steep slope was the main reason for the high risk of loss in this area. The overall situation of nitrogen loss risk in maize planting area was higher than that in cassava planting area, which further validated the reasons for TN over-proof in the two study areas. The Hulan River was facing more serious standard exceeding of TN.

The occupation ratios for the maize platting in risk zones in the north were 54% (2~3), 38% (4~5) and 41% (>5). The overall trend showed the percentage of maize decreased with the risk level went up. The similar rules occurred in the soybean circumstances while rice cover percentage went from 2% to 8% in high-risk area. Considering the near river location of high-risk zone, the relatively easily washed-out characters would reveal those sites far away from the riverbank could be better choices for rice cultivation. Other kinds of crops had the potential to give priority for maize or soybean, more or less “Returning farmland to grassland” policy could be a propriate direction for reducing soil nitrogen loss risk.

As for the south research area, rice took the biggest ratio and followed by forest and other cultivated land. Rice experienced the same trend as the north maize, the occupation percentage were 40% (2~3), 32% (3~4) and 24% (4~5). The risk zone of each level contained around 2% of cassava and 4% of maize. Other crops covered more with the risk level ascended, that would lead to probabilities of alternatives for cassava, rice or maize planting in order to secure environment and economic purposes. Forest played positive role for controlling the risk, so “Returning farmland to forest” policy would be feasible. 

#### 3.4.2. Risk Assessment of Soil Phosphorus Loss

Figure 14 showed the risk distribution of soil phosphorus loss. Most of the region was at high-risk loss status in maize planting area. The most serious area was located on the north bank from the middle reaches. The soil phosphorus index in cassava planting area was mainly in the range of <3 and 3~4 representing low and medium risk. Two low risk regions were situated on upper reaches and the estuary district, north of the river basin. 

Maize contributed the most for phosphorus loss risk in the north. Rice and soybean devoted 4% and 3% in high-risk zone. That indicated similar regularities with the nitrogen characters. Regarding the south research area, the rice took over 23% to 31% in each risk level, while maize and cassava occupied about 3%. Unlike the nitrogen characters, rice accounted more with the risk level exacerbating. Maize and cassava stayed smooth in risk zone performance. To sum up, there was a certainly need to balance nitrogen and phosphorus effects which imposed by the rice planting layout. For both area, other cultivated land should be considered the conversion to patterns such as forest, grassland or crops including cassava or maize.

## 4. Conclusions

In this study, two typical fuel ethanol raw material planting areas in the South and North were selected to carry out the distribution of water quality indexes and soil nitrogen and phosphorus forms in the river basin. Combined with the loss risk assessment based on planting structures of crops, the agricultural non-point source pollution characteristics in the two study areas were discussed.

TN pollution was serious in Harbin section of the Hulan River, and the TP concentrations of some sample points could not meet the standard of water functional area. The water quality indexes of the Yujiang River were significantly better. The river pollution in maize planting area was greatly affected by TN, TP, Ex-P and Fe/Al-P in soil, while soil TN and NO_3_^−^-N were the main factors influencing the water pollution of the Yujiang river.The spatial distribution of nitrogen and phosphorus loss risk assessment revealed that most of the maize planting areas were at medium or high nitrogen loss risk, and the overall risk was higher than that of cassava planting area in the Guiping section. For planting suggestions of both areas, other cultivated land should be considered the conversion to patterns such as forest, grassland or crops including cassava, maize or soybeans. Rice needed more deep discusses to balance the nitrogen and phosphorus loss.

In summary, the main raw material planting area of bioethanol fuel the soil nitrogen and phosphorus loss risk of cassava planting in the south was significantly lower than that of maize planting area in the north. It had less impact on the water quality in the river basin, and was suitable for vigorously developing for fuel ethanol. Maize planting area should strengthen land management and non-point source pollution control, and pay more attention to the aggravation of nitrogen and phosphorus pollution caused by the expansion of maize planting area with fuel ethanol policy variations. For the areas at high-risk status, it was necessary to take the lead in improving the utilization and treatment of chemical fertilizer. Under the guarantee of fuel-ethanol raw material demand, it was the priority to encourage the planting of crops with low fertilizer usage and high economic value. For the long-term planning, it was essential to promote sustainable agriculture such as “bio-ethanol raw material cultivation-ethanol preparation-feed processing-livestock and poultry breeding-manure composting-organic fertilizer returning”. This study would provide scientific support and management suggestions for the assessment of the national fuel ethanol policy impacts on the water environment of the raw material planting area.

## Figures and Tables

**Figure 1 ijerph-19-01394-f001:**
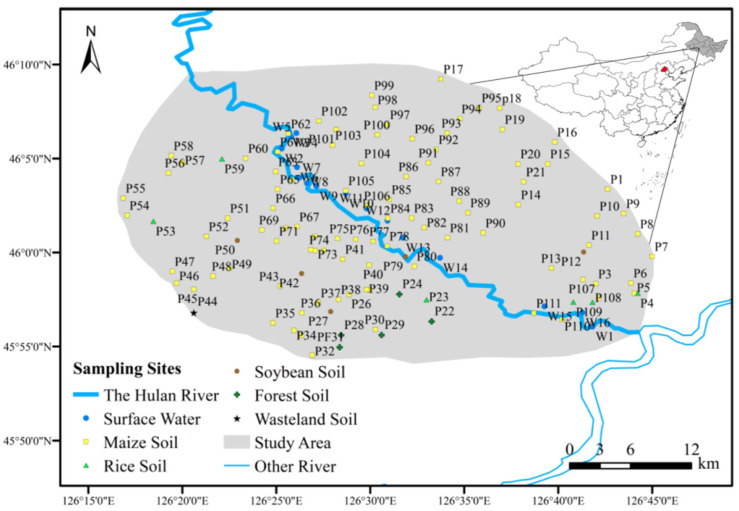
Overview of the maize planting area and sampling points’ distribution.

**Figure 2 ijerph-19-01394-f002:**
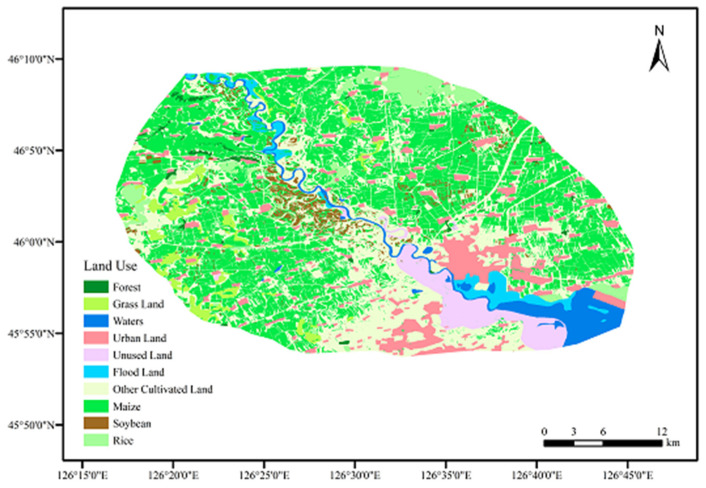
Land use types in maize planting area.

**Figure 3 ijerph-19-01394-f003:**
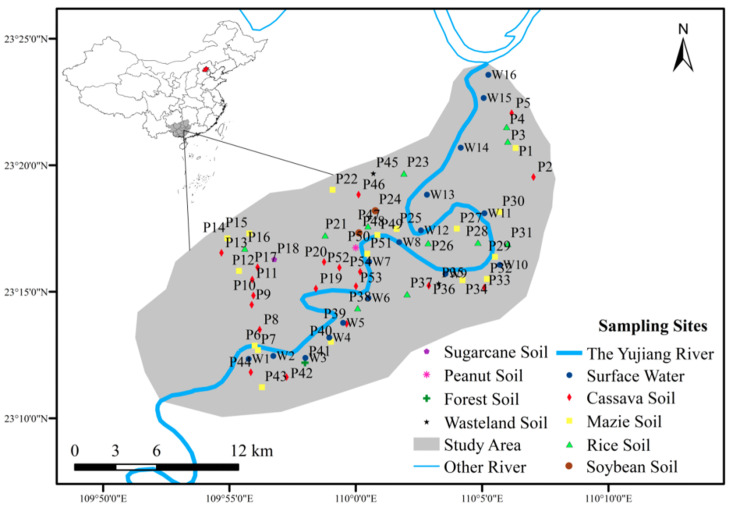
Overview of the cassava planting area and sampling points’ distribution.

**Figure 4 ijerph-19-01394-f004:**
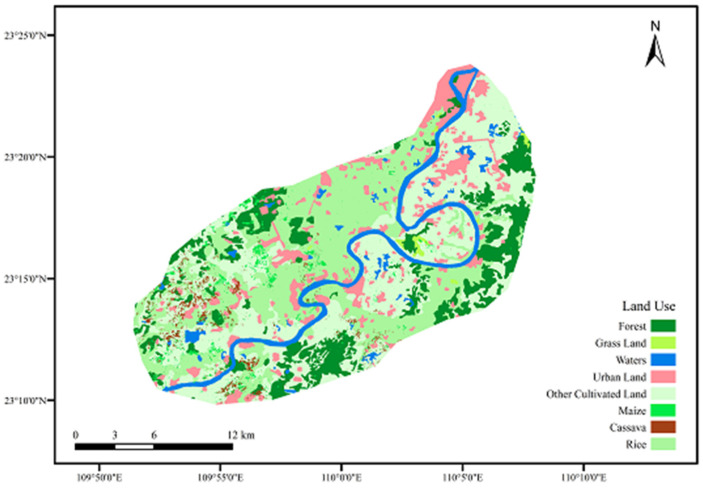
Land use types in cassava planting area.

**Figure 5 ijerph-19-01394-f005:**
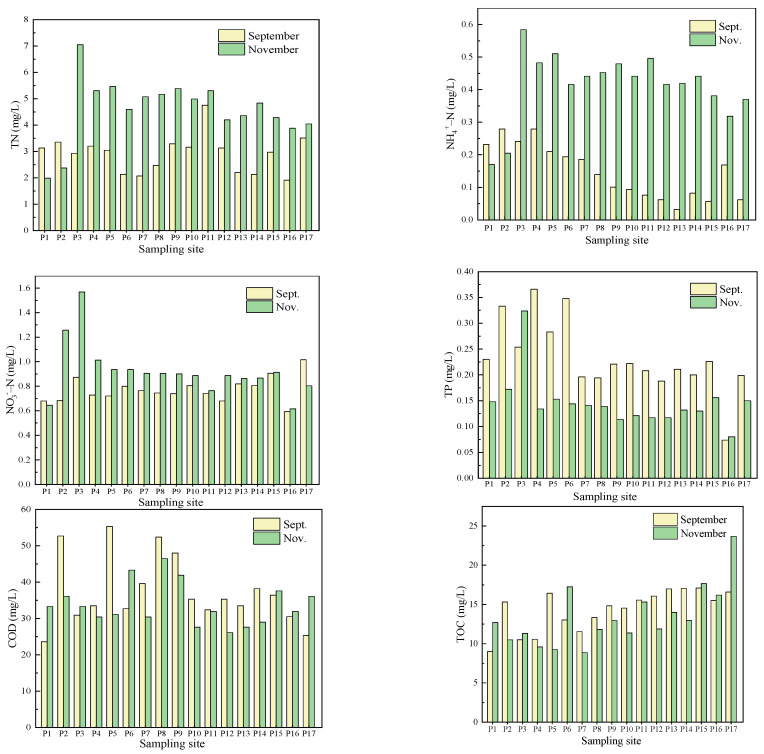
Water quality index concentrations of Hulan River.

**Figure 6 ijerph-19-01394-f006:**
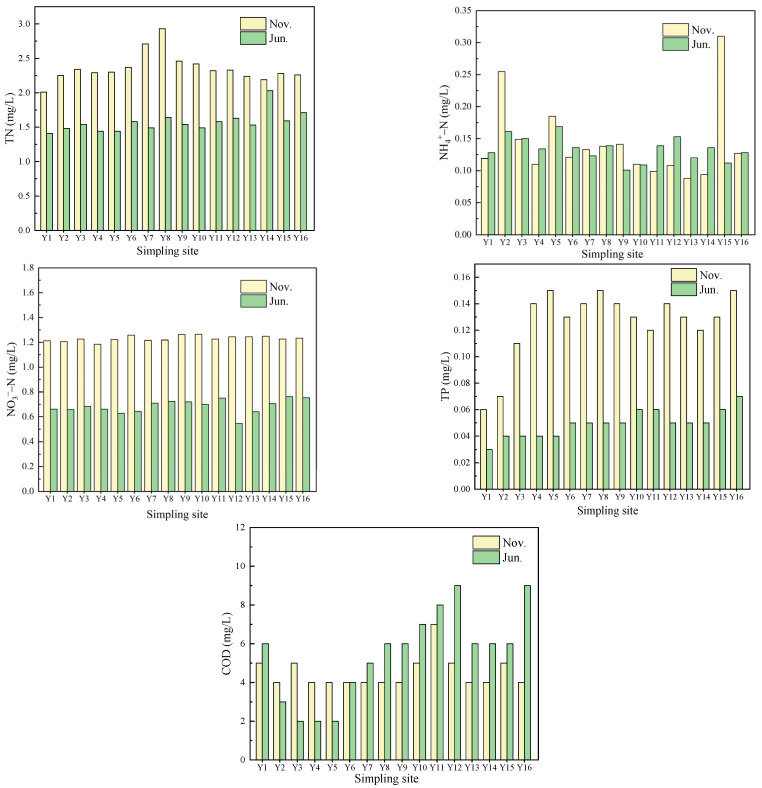
Water quality index concentrations of Yujiang River.

**Figure 7 ijerph-19-01394-f007:**
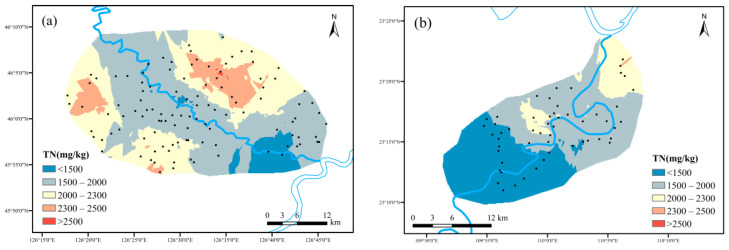
Spatial distribution characteristics of TN: ((**a**) maize planting area; (**b**) cassava planting area).

**Figure 8 ijerph-19-01394-f008:**
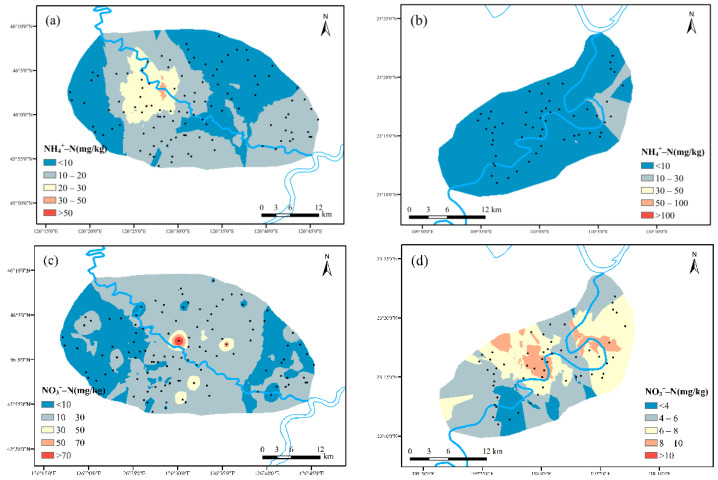
Spatial distribution characteristics of NH_4_^+^-N and NO_3_^−^-N: ((**a**,**c**) maize planting area; (**b**,**d**) cassava planting area).

**Figure 9 ijerph-19-01394-f009:**
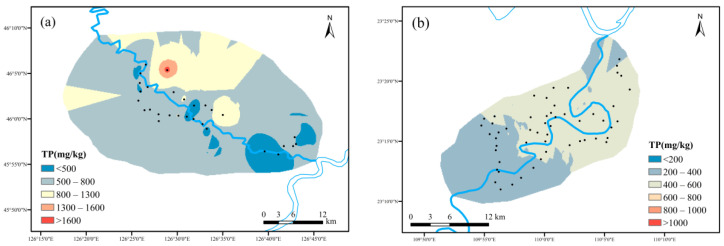
Spatial distribution characteristics of TP: ((**a**) maize planting area; (**b**) cassava planting area).

**Figure 10 ijerph-19-01394-f010:**
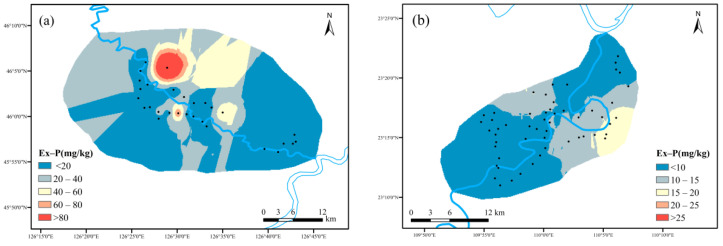
Spatial distribution characteristics of Ex-P: ((**a**) maize planting area; (**b**) cassava planting area).

**Figure 11 ijerph-19-01394-f011:**
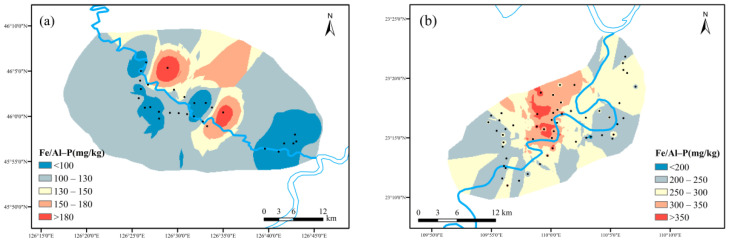
Spatial distribution characteristics of Fe/Al-P: ((**a**) maize planting area; (**b**) cassava planting area).

**Figure 12 ijerph-19-01394-f012:**
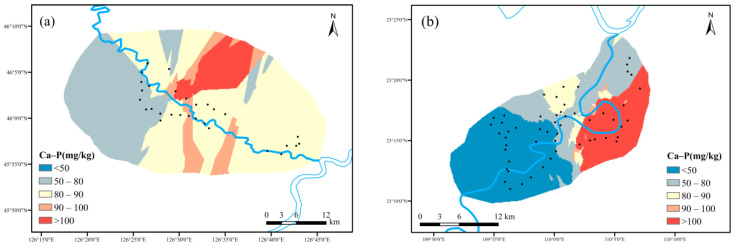
Spatial distribution characteristics of Ca-P: ((**a**) maize planting area; (**b**) cassava planting area).

**Figure 13 ijerph-19-01394-f013:**
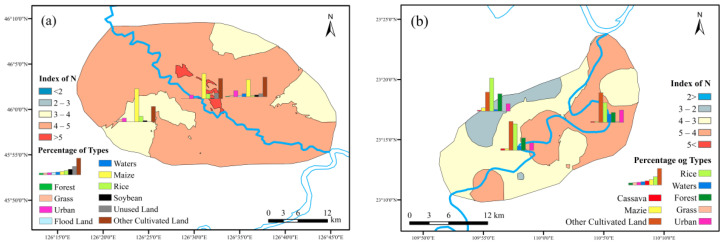
Soil nitrogen loss risk zones: ((**a**) maize planting area; (**b**) cassava planting area).

**Figure 14 ijerph-19-01394-f014:**
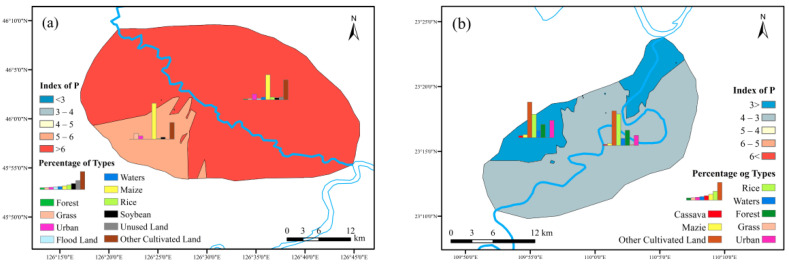
Soil phosphorus loss risk zones: ((**a**) maize planting area; (**b**) cassava planting area).

**Table 1 ijerph-19-01394-t001:** Determination methods for water and soil nitrogen and phosphorus forms.

Type	Content	Method/Instrument
Determination of water elements	TN	Alkaline potassium persulfate oxidation UV spectrophotometry
	NH_4_^+^-N	Nessler reagent spectrophotometry
	NO_3_^−^-N	Ultraviolet spectrophotometry
	TP	Ammonium molybdate spectrophotometry
Determination of soil nitrogen forms	NH_4_^+^-N	Nessler reagent spectrophotometry
	NO_3_^−^-N	Ultraviolet spectrophotometry
	TN	Element analyzer
Determination of phosphorus forms	TP	SMT UV Spectrophotometry
	Ex-P	
	Fe/Al-P	
	Ca-P	SMT visible spectrophotometry

**Table 2 ijerph-19-01394-t002:** Risk assessment system of nitrogen loss (mg kg^−1^).

Factor	Weight	Lower 1	Low 2	Medium 4	High 8	Higher 10
TN	0.4	<1200	1200~1500	1500~1800	1800~2500	>2500
Application rate	0.9	0~100	100~200	200~400	400~600	>600
Method	0.8	Buried	Scatter	Surface	Surface	Surface
Period	0.7	Early spring	Summer	Late summer	Summer and Fall	Summer
**Factor**	**Weight**	**Lower** **0.6**	**Low** **0.7**	**Medium** **0.8**	**High** **0.9**	**Higher** **1.0**
Soil erosion	1	<2	2~10	10~25	25~50	>50
Distance from river	1	<3	2~3	1~2	0.5~1	<0.5

**Table 3 ijerph-19-01394-t003:** Risk assessment system of phosphorus loss (mg kg^−1^).

Factor	Weight	Lower 1	Low 2	Medium 4	High 8	Higher 10
TP	0.4	<500	500~700	700~900	900~1000	>1000
Application rate	0.9	0~30	30~100	100~150	150~200	>200
Method	0.8	Buried	Scatter	Surface (after planting)	Surface (within a month)	Surface (after a month)
Period	0.7	Early spring	Summer	Late summer	Summer and Fall	Summer
**Factor**	**Weight**	**Lower** **0.6**	**Low** **0.7**	**Medium** **0.8**	**High** **0.9**	**Higher** **1.0**
Soil erosion	1	<2	2~10	10~25	25~50	>50
Distance from river	1	<3	2~3	1~2	0.5~1	<0.5

**Table 4 ijerph-19-01394-t004:** Principal component analysis results of water quality (Hulan River).

	Principal Component 1	Principal Component 2	Principal Component 3
TN	0.950	0.264	−0.046
NH_4_^+^-N	0.957	0.174	−0.066
NO_3_^−^-N	0.192	0.956	−0.053
TP	0.161	0.922	0.041
TOC	−0.095	−0.304	0.821
COD	0.080	0.116	0.694
Turbidity	0.576	−0.134	−0.045
Characteristic value	2.910	1.608	1.183
Variance (%)	41.579	22.978	16.893

**Table 5 ijerph-19-01394-t005:** Principal component analysis results of water quality (Yujiang River).

	Principal Component 1	Principal Component 2	Principal Component 3
TN	0.026	0.832	−0.125
NH_4_^+^-N	−0.433	0.003	0.848
NO_3_^−^-N	0.832	−0.112	−0.018
TP	0.503	0.718	0.036
PH	0.903	0.237	−0.049
COD	0.069	−0.610	−0.121
Turbidity	0.524	−0.055	0.798
Characteristic value	2.508	1.412	1.350
Variance (%)	35.832	20.171	19.286

**Table 6 ijerph-19-01394-t006:** Nitrogen and phosphorus application rates of different crops in Harbin and Guiping.

	Type	Nitrogenous Fertilizer (kg (mu·year)^−1^)	Phosphate Fertilizer (kg (mu·year)^−1^)
Maize planting area	Maize	7.32	0.53
	Soybean	1.48	0.02
	Rice	23.54	2.86
Cassava planting area	Maize	12.37	1.14
	Cassava	0.8	0.1
	Rice	8.21	0.39

**Table 7 ijerph-19-01394-t007:** Soil erodible factors in maize planting area and cassava planting area.

Soil Type	Soil Erodible Factors K (t (hm^2^·a)^−1^)
Meadow black soil	0.2489
Meadow chernozem	0.2501
Acid purple soil	0.0196
Yellow latosolic red soil	0.0065
Paddy soil	0.0185

**Table 8 ijerph-19-01394-t008:** Risk assessment and rating method of soil nitrogen and phosphorus loss.

Risk Level	Lower	Low	Medium	High
Nitrogen index	<1	1~2	2~5	>5
Phosphorus index	<1	1~3	3~6	>6

## Data Availability

The data presented in this study are available on request from the corresponding author.

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
