# Peer review of "Distribution Characteristics and Risk Assessment of Agricultural Land Use Non-Point Source Pollution in Typical Biofuel Ethanol Planting Areas"

_ijerph, 2022, doi:10.3390/ijerph19031394_

Round 1
Reviewer 1 Report
ijerph-1483174
L20: Was this threat more severe to the cassava crop?
L21: Does this river water pollution apply to the maize growing area?
L24: The authors should explain why they claim this.
L29: Please provide other keywords (do not repeat the same words that are already used in the title of the article)
L36-45: I do not share the authors' optimism. Combustion of biofuels does result in CO2 emissions. Therefore, biofuels are not an "ideal" substitute for non-renewable fuels. It is also difficult to see the large-scale monoculture of crops grown for biofuel production as environmentally friendly. E.g. do these crops give a zero balance for CO2, and what about the issue of using chemical protection agents and fertilizers for such crops, or the issue of genetic modification?
L106-107: Wasn't it too short a duration between the times of sampling for testing?
Figs 1 and 3 are not very clear. I cannot see the numbering of the river water sampling sites. Please enlarge the legend.
L117 and L137: Please provide the % share of these crops.
L128: Why was testing done in only one season for the cassava area (although two sampling dates were included in Fig. 6)? This sampling date was also does not correspond to the sampling of the maize growing area. Could these differences in hydrological regime affect the comparability of calculated indices?
What methods were used to determine water quality indicators?
Tables 2, 3, 8: In the description of the tables, please provide the source of these data.
L167: Please provide the numerical values of these factors
L238: Probably TP and not TN.
In the current form, the description of results in subsections 3.1.1. and 3.1.2. is unclear and should be reworded according to the following scheme: characterization of the range of concentrations; general trend of changes in concentrations along the course of the rivers; differences of these trends depending on the season; comparison of the two rivers with each other; reference of the pollution level, also in relation to other standards (not only the national standard). Why is the temporal and spatial variability of the other water quality indicators not discussed? Also, it is not known (due to the poor quality of Figs 1 and 3) whether the increase in N and P pollution was consistent with the course of the river (i.e. increasing from springs to the river)?
How does N-NO3- affect the release of TP from river sediments?
How can CO2 be combined with TOC analysis?
Figs 7-13: Please enlarge the font of the legend.
L356, L395: Please add these standards to the literature list.
L358-360; L397-399: Are these divisions consistent with the previously mentioned national standards?
The authors do discuss their results with the literature at all.
They refer to no studies on N and P release from soils in reference areas, e.g. forests or protected regions (e.g. landscape or national parks) from the two regions.
L545-547: In my opinion this is not "ecological agriculture" but "sustainable agriculture ".
Reviewer 2 Report
The authors have conducted a valuable study and have presented the work well. Except for a few minor issues I do not have any major suggestions:
1. The quality of Figure 1 and Figure 3 is very poor. It difficult to understand what is what in the Figure !
2. Should the section be 2 instead of 1 in '1. Materials and methods' !
3. I am not sure why all of a sudden a section starts with '(2) Cassava planting area (Guiping section of Yujiang River Basin)'. May be the authors intended to break up the '2.1. Overview of the study area' in two sections. Please make the appropriate changes.
4. Again not sure why its named section 1.2 in '1.2. Experimental method'. Should this be 2.2 ? Please review the numbers in every section carefully.
5. Sections such as 'Estimation of source factors' etc. is not properly identified. Please fix all the issues with sections.
6. 'The TN distribution map was 353 calculated by Kriging interpolation method by GIS.' - why was this interpolation method used ? Why not something else ? What was the grid size ? Also - what GIS software was used ?
7. Can you extend the use of this study to other areas in China or the world - or are the results only valid for this study region only ? Applicability of the findings to other regions increase the usefulness of the study.
8. References - 10 is in red. After 11 - all references are double numbered.
Round 2
Reviewer 1 Report
The manuscript has been improved as the reviewer's suggestion and it is acceptable now